# An Engineered Thermostable Laccase with Great Ability to Decolorize and Detoxify Malachite Green

**DOI:** 10.3390/ijms222111755

**Published:** 2021-10-29

**Authors:** Guotao Mao, Kai Wang, Fangyuan Wang, Hao Li, Hongsen Zhang, Hui Xie, Zhimin Wang, Fengqin Wang, Andong Song

**Affiliations:** 1Department of Microbiology, College of Life Sciences, Henan Agricultural University, Zhengzhou 450002, China; maoguotao@henau.edu.cn (G.M.); kaiwang@stu.henau.edu.cn (K.W.); wangfangyuan@ct-bio.com (F.W.); haoli2021@stu.henau.edu.cn (H.L.); hszhang@henau.edu.cn (H.Z.); xiehuiliving421@vip.sina.com (H.X.); w_fengqin@henau.edu.cn (F.W.); 2The Key Laboratory of Enzyme Engineering of Agricultural Microbiology, Ministry of Agriculture, Henan Agricultural University, Zhengzhou 450002, China; 3Department of Applied Chemistry, College of Science, Henan Agricultural University, Zhengzhou 450002, China; gary1451@iccas.ac.cn

**Keywords:** thermostable laccase, malachite green, decolorization, detoxification

## Abstract

Laccases can catalyze the remediation of hazardous synthetic dyes in an eco-friendly manner, and thermostable laccases are advantageous to treat high-temperature dyeing wastewater. A novel laccase from *Geothermobacter hydrogeniphilus* (Ghlac) was cloned and expressed in *Escherichia coli*. Ghlac containing 263 residues was characterized as a functional laccase of the DUF152 family. By structural and biochemical analyses, the conserved residues H78, C119, and H136 were identified to bind with one copper atom to fulfill the laccase activity. In order to make it more suitable for industrial use, Ghlac variant Mut2 with enhanced thermostability was designed. The half-lives of Mut2 at 50 °C and 60 °C were 80.6 h and 9.8 h, respectively. Mut2 was stable at pH values ranging from 4.0 to 8.0 and showed a high tolerance for organic solvents such as ethanol, acetone, and dimethyl sulfoxide. In addition, Mut2 decolorized approximately 100% of 100 mg/L of malachite green dye in 3 h at 70 °C. Furthermore, Mut2 eliminated the toxicity of malachite green to bacteria and *Zea mays*. In summary, the thermostable laccase Ghlac Mut2 could effectively decolorize and detoxify malachite green at high temperatures, showing great potential to remediate the dyeing wastewater.

## 1. Introduction

Laccases (benzenediol: oxygen oxidoreductase, EC 1.10.3.2), a member of polyphenol oxidases, can oxidize a wide range of phenolic and nonphenolic compounds [1]. Laccases typically contain four copper atoms in their active center [1]. Type 1 copper, which exhibits strong electronic absorbance at 610 nm, can abstract one electron from the substrate. During the subsequent electron transferring, oxygen is reduced to water at the trinuclear center formed by Type 2 and 3 copper [2]. Thanks to their broad substrate specificity and the green reaction only requiring oxygen and releasing water as the sole by-product, laccases have been applied for industrial use such as delignification to improve biomass saccharification [3], biobleaching [4], degradation of environmental pollutants [5], and decolorization and detoxification of dyes [6,7].

Laccases are widely distributed in fungi, plants, insects, and bacteria. Bacterial laccases exhibit rather low redox potential about 400 mV as compared with fungal laccases with higher redox potential between 470 and 810 mV [8]. Because of the high redox potential, the laccases from fungi have been the focus of research, and their applications have been extensively exploited [9]. However, the long production cycle, poor thermostability, and low tolerance for the alkaline condition hinder the practical application of fungal laccases [9]. Recently, bacterial laccases have been found to possess advantageous characteristics, including good stability under high temperature and alkaline conditions [2,5,10]. Besides, with the help of a redox mediator, bacterial laccases could gain the ability to degrade the recalcitrant substrates with higher redox potential than that of bacterial laccases [11,12]. Therefore, bacterial laccases could be promising alternatives to fungal laccases for some specific industrial applications.

Malachite green (MG), a triphenylmethane dye, is extensively used in the textile dyeing industry [13]. As a recalcitrant chemical with teratogenic, carcinogenic, and mutagenic effects, the large amount of discharged MG persistently threatens the environment and public health [13,14]. The fungal laccases from *Trametes* sp. 48424, *Cerrena* sp., and *T. asperellum*, as well as bacterial laccases from *Bacillus pumilus*, *Bacillus* sp. FNT, and *Sulfitobacter indolifex*, were demonstrated to decolorize MG under mesothermal conditions [15,16,17,18,19]. However, the temperature of wastewater released from the dyeing process is always above 50 °C [20], and a higher temperature usually means higher decolorization velocity [21]. In order to avoid the extra cooling process to reduce the cost and take full advantage of the high temperature of the dyeing wastewater to fulfill the maximum decolorization rate in a short period, laccases with high optimal temperatures and excellent thermostability are required.

The DUF152 laccases, a new subfamily of the bacterial laccases, were characterized in 2006. The molecular weights (about 30 kDa) and amino acid sequences of the DUF152 laccases are quite different from those of typical laccases (50–130 kDa) [22]. The isolated DUF152 laccases RL5 from a metagenome expression library of the bovine rumen, Tfu1114 from *Thermobifida fusca*, and LaclK from *Kurthia huakuii* had high optimal temperatures (above 60 °C) and showed excellent thermostability [22,23,24]. Their potential to decolorize different dyes such as poly-R 478, ethyl violet, and methylene blue was demonstrated [22,24]. Besides, thanks to its good thermostability, Tfu1114 was incorporated into the cellulosome, significantly enhancing the ability to hydrolyze the unpretreated wheat straw [25]. Therefore, the DUF152 laccases showed great potential to treat high-temperature dyeing wastewater. Herein, a novel member of DUF152 laccases, Ghlac, was characterized from *Geothermobacter hydrogeniphilus*, and its putative copper binding site was identified. In addition, Ghlac variant Mut2 with improved thermostability was engineered, and its capability of decolorizing MG at high temperatures was assessed. After Mut2 treatment, the toxicity of MG to bacteria and plants was evaluated to promote its practical application.

## 2. Results and Discussion

### 2.1. Sequence Analysis, Expression, Purification, and Mutation of Ghlac

The open reading frame of *Ghlac* encoding an uncharacterized protein containing the consensus sequences of DUF152 laccases was found in the thermophile *G. hydrogeniphilus*. Ghlac contains 263 residues with a theoretical molecular weight of 29.0 kDa. Multiple sequence alignment showed that Ghlac shares 22.6%, 30.2%, and 34.0% identities to LaclK, RL5, and Tfu1114, respectively (Figure 1A). The putative copper binding sites (N41, H78, C119, and H136) were conserved in Ghlac [22]. The structure model indicated that Ghlac has a similar structural fold to the DUF152 member GsYlmD (Figure 1B).

As aforementioned, we suggested that Ghlac is a putative functional laccase. To verify this, Ghlac was cloned, expressed, and purified using Ni-NTA chromatography (Figure 1C). The molecular weight of the purified homogeneous Ghlac corresponded to the predicted size (Figure 1C). The activity assay showed that Ghlac could oxidize 2,2′-azino-bis(3-ethylbenzthiazoline)-6-sulfonate (ABTS), the typical substrate of laccases. *K*_m_ and *k*_cat_ of Ghlac were 1.3 mM and 125.7 min^−1^, respectively (Figure 2A), which are comparable to those of the DUF152 laccases Tfu1114 and LaclK and the typical laccase pLac*_Si_* from *S. indolifex* [18,23,24].

The half-life (t_1/2_) of Ghlac wild type (WT) at 50 °C was less than 24 h (Figure 3D), which could hardly satisfy the requirement for industrial application. In order to improve the thermostability, Ghlac variants Mut1, Mut2, and Mut3 were designed by PROSS and characterized (Appendix A) [26]. Among these variants, Mut2 showed increased thermostability, whereas Mut1 and Mut3 exhibited decreased activity and thermostability (Figure 3 and Appendix A). Therefore, Mut2 was further studied.

### 2.2. Effects of pH and Temperature on the Activity and Stability of Ghlac

The optimal pH for Ghlac WT and Mut2 against ABTS was 4.0 (Figure 3A), which is in accordance with the acidic pH preference of fungal and bacterial laccases [9,19]. Ghlac lost more than 60% of its original activity after incubation at pH 3.0 for 6 h, while Mut2 retained more than 95% of the original activity over the pH range of 4.0 to 8.0 (Figure 3C), similar to the characterized DUF152 laccases [22,23,24]. The bacterial laccases from *B. stratosphericus*, γ-proteobacterium, and a marine microbial metagenomic library showed high tolerance for alkaline conditions, which is an advantageous property of laccases from bacteria [27,28,29]. By contrast, most of the laccases from fungi are unstable under alkaline conditions [5,30].

Ghlac WT and Mut2 showed their maximal activity at 60 °C (Figure 3B). Their thermostability was tested to evaluate the potential for industrial application. Mut2 retained 100% of its original activity after incubation at 50 °C for 6 h, whereas Ghlac WT lost 20% of its activity. Furthermore, t_1/2_ of Ghlac was calculated (Figure 3D). t_1/2_ of Mut2 at 50 °C was 80.6 h, 3.7 times longer than that of WT. t_1/2_ of Mut2 at 60 °C was increased to 9.6 h, compared with that of WT. Additionally, T_m_ of Mut2 was 6.9 °C higher than that of WT (Figure 4A), which is consistent with the results of t_1/2_ measurements. Most of the fungal laccases could not retain 50% of the original activity after incubation at 60 °C for 6 h [30]. rLac from *Klebsiella pneumoniae* only retained 60% and 50% of the activity for 5 h at 50 °C and 60 °C, respectively [31]. The spore-coat laccase FNTL from *Bacillus* sp. lost 80% of the activity at 60 °C for 5 h [19]. Therefore, Mut2 is highly thermostable. Based on the principle of PROSS and the structural mapping of the mutated residues of Mut2 (Appendix A), the introduced mutations increased the surface polarity (such as S58E and G197E) and rigidified the flexible elements (such as A35P and V68P), significantly enhancing the thermostability of Mut2 [26,32,33].

The determined *K*_m_ and *k*_cat_ of Mut2 were comparable to those of Ghlac WT (Figure 2A). Mut2 was able to oxidize ABTS, 2,6-dimethoxyphenol (DMP), and guaiacol, and the substrate preference decreased in the order of DMP > guaiacol > ABTS (Figure 2B). However, syringaldazine (SGZ) could not be catalyzed by Mut2.

Therefore, Mut2 is a functional polyphenol oxidase of the DUF152 family, although it was cloned from an anaerobe. Berini et al. also reported a functional laccase from the anaerobe *Geobacter metallireducens* and discussed the possible mechanism using small amounts of oxygen in the environment or other chemical (such as N_2_O) as the electron acceptor [34]. The physiological role of *Ghlac* in *G. hydrogeniphilus* needs to be further studied by thorough in vivo experiments. Overall, the excellent thermostability makes Mut2 a potential catalyst for industrial application.

### 2.3. Identification of the Putative Copper Binding Site

The typical laccase contains four copper atoms, and the copper is required to transfer electrons during catalysis [5]. When the concentration of Cu^2+^ increased, the activity of Mut2 also increased accordingly, indicating that Mut2 is Cu^2+^-dependent (Figure 4C). Moreover, the Thermofluor assay showed that T_m_ of Mut2 was increased from 56.4 °C to 60.9 °C in the presence of Cu^2+^ (Figure 4A), implying that Cu^2+^ helps to stabilize Mut2 by interaction [35]. However, the UV/visible spectra of Mut2 (Figure 4B), Tfu1114, and LaclK lacked the absorption peak around 610 nm, a characteristic of the typical laccases [23,24], showing that DUF152 laccases might interact with the copper in a different way. Furthermore, ICP-MS detected 1.1 ± 0.1 mol/mol of copper in Mut2 (Figure 4F), which corresponds to the copper content of Tfu1114 and LaclK, but differs from that of RL5 (4.0 ± 0.2 mol/mol) [22,23,24]. Among the 12 identified residues forming the putative copper binding sites in RL5 [22], only N41, H78, C119, and H136 are conserved in the characterized DUF152 laccases (Figure 1A). Thus, could the four conserved residues constitute the putative copper binding site of Mut2?

Structural analysis showed that N41 of Mut2 is far from the cluster formed by H78, C119, and H136 (Figure 1B), and the cluster is bound with one Zn^2+^ in the crystal structure of GsYlmd (Figure 4D). Hence, variants H78A, C119A, H136A, and 3A were constructed to confirm whether the putative binding site is formed by H78, C119, and H136. Variants H78A, C119A, and H136A showed significantly decreased laccase activity, while variant 3A almost abolished its activity (Figure 4E). Meanwhile, no significant difference was detected between the T_m_ values of Mut2 and the variant 3A (Figure 4A), indicating that the mutation of H78, C119, and H136 did not change the native structure of Mut2. Furthermore, no copper was detected in the variant 3A (Figure 4F). Correspondingly, the addition of Cu^2+^ did not increase T_m_ of 3A (Figure 4A).

Taken together, the conserved residues H78, C119, and H136 constitute the putative copper binding site of Mut2 and bind with one copper atom to exert the laccase activity. However, the substrate binding site and the catalytic mechanism of Mut2 are yet to be elucidated by crystal structures of Ghlac complexed with substrates.

### 2.4. Effects of Metal Ions and Organic Solvents on the Activity of Mut2

As shown in Figure 5A, Mut2 maintained over 82% residual activity in the presence of 10 mM Na^+^, Mg^2+^, Ca^2+^, Mn^2+^, and Ni^2+^. Although the activity of Mut2 was not strongly affected by 1 mM Zn^2+^ and Ba^2+^, it was dramatically reduced to 47% and 0, respectively, when the metal ions concentration increased to 10 mM. This inhibitory effects of high-concentration Zn^2+^ or Ba^2+^ were generally reported in laccases such as BaCotA, PvL, SN4LAC, and MSKLAC [27,36,37,38]. As the essential factor for the activity of Mut2, 10 mM Cu^2+^ did not show any inhibitory effect on Mut2 (Figure 4C). With the addition of 1 mM chelating agent EDTA, Mut2 completely lost its activity, indicating the importance of Cu^2+^ during catalysis.

The effects of organic solvents on the activity of Mut2 were also assessed (Figure 5B). Mut2 retained more than 90% and 80% of its original activity in the presence of 10% and 20% methanol, ethanol, acetone, and isopropanol, respectively. In the presence of 10% dimethyl sulfoxide, the activity of Mut2 was boosted to 120%. This kind of boosted laccase activity was also reported in laccases from *Cerrena* sp. RSD1, *T. versicolor*, and a marine metagenomic library [39,40], whereas laccases from *B. stratosphericus*, *Ganoderma lucidum*, and *Sporothrix carnis* were strongly inhibited by 10% dimethyl sulfoxide [27,41,42]. Moreover, Mut2 could retain 90% of its activity in the presence of 10% formaldehyde, which is a powerful protein denaturant, and could strongly inhibit the laccase from *B. amyloliquefaciens* [43]. These results indicated that Mut2 is tolerant to the organic solvents.

Overall, Mut2 is quite resistant to common environmental pollutants, including metal ions and organic solvents.

### 2.5. MG Decolorization Catalyzed by Mut2

The dark color and strong toxicity of MG discharged from the dyeing industry severely threaten the environment and human health [13]. However, conventional physical and chemical treatment processes usually produce secondary sludge and hazardous byproducts, resulting in serious environmental pollution [44,45]. Laccase, as a green catalyst, has drawn increasing attention, being a potential alternative to tackle dye pollution in an environmentally friendly manner [27,45,46].

Thanks to the reasonably good thermostability of Mut2 and the high temperature of the dyeing wastewater [47,48], the capability of MG decolorization by Mut2 was investigated at high temperatures. Mut2 alone decolorized only 10% of 100 mg/L of MG at 60 °C in 2 h. Redox mediators have been used to improve dye decolorization [11]. Among the tested mediators, ABTS was the most effective one (Figure 6A). With the help of 0.1 mM ABTS, the MG decolorization rate by Mut2 was improved to 90% (Figure 6B). The improvement by ABTS is in accordance with the study of laccases BaCotA and rLAC [27,46]. The optimal pH and temperature for MG decolorization were 3.5–4.0 and 70 °C, respectively (Figure 6C,D). The decolorization rate was more than 90% in the temperature range of 60–75 °C, and the highest rate of 98% was reached at 70 °C in 2 h. Decolorization was almost completed by Mut2 in 3 h, whereas only 78% of MG was decolorized by Ghlac WT, showing the advantage of thermostable laccases to treat high-temperature dyeing wastewater (Figure 6E and Appendix A). The disappearance of the absorption peak at 617 nm indicated the destruction of the conjugated chromophore structure of MG, which showed that Ghlac Mut2 may decolorize MG by the oxidation and cleavage of the chromophore of MG in a similar way to the typical laccases [15,16].

The textile effluent usually has a relatively high temperature, above 50 °C and even up to 70–80 °C [20,47], and a high temperature would benefit the decolorization velocity [21]. In order to avoid the extra cooling process and maximize the decolorization rate, more and more studies focus on MG decolorization by laccases at high temperatures (Table 1). LaclK of the DUF152 family was highly thermostable, but its ability to decolorize MG was poor [24]. Although BaCotA, CueO-p, rLAC, and rLac could effectively decolorize MG, the relatively low thermostability would result in incomplete decolorization, even with an extended incubation time [27,31,46,49]. Thus, the remaining MG would still threaten public health. Compared with these laccases, the identified Mut2 herein is superior in both thermostability and the completeness of decolorization and, consequently, more suitable for industrial application.

### 2.6. MG Detoxification Catalyzed by Mut2

The toxicity of MG before and after Mut2 treatment was estimated to evaluate whether Mut2 could detoxify MG.

As Figure 7A,B illustrated, *Escherichia coli* and *B. subtilis* did not grow in LB containing untreated MG, indicating that MG is toxic to bacteria. As expected, in both the control group and the treated MG group, bacteria grew in almost the same manner, indicating that the toxicity of MG to bacteria was eliminated by Mut2. A phytotoxicity test of MG before and after Mut2 treatment was also carried out with *Zea mays* by recording seed germination and the elongation of radical and plumule (Figure 7C–E). Compared with the control group, MG inhibited the germination of *Z. mays* seeds by 55%, and the length of radical and plumule was only about 38% of that in the control group. In the treated MG group, the growth of *Z. mays* seeds showed no difference from that in the control group (Figure 7C–E). However, MG treated by the laccase LacA from *Cerrena* sp. still inhibited the root elongation of *Nicotiana tabacum* and *Lactuca sativa* [15]. These results revealed that Mut2 could eliminate the toxicity of MG to bacteria and *Z. mays*.

## 3. Materials and Methods

### 3.1. Materials

The genes encoding Ghlac WT and variants were ordered from Genscript. ABTS, DMP, guaiacol, SGZ, 1-hydroxybenzotriazole (HBT), acetosyringone (ASG), and Sypro Orange were obtained from Sigma-Aldrich (Saint Louis, MO, USA). Methyl syringate (MeS) and violuric acid (VA) were purchased from BioRuler (Danbury, CT, USA). Malachite Green chloride (>98%) was purchased from Shanghai yuanye Bio-Technology Co., Ltd. (Shanghai, China). All other chemicals and reagents were of analytical grade.

### 3.2. Cloning, Expression, and Purification of Ghlac

The gene encoding Ghlac WT (NCBI accession No.: ORJ60343.1, https://www.ncbi.nlm.nih.gov/protein/ORJ60343.1, accessed on 15 June 2019) was codon-optimized, synthesized, and cloned in the pET-28a (+) vector using the *Nco* I and *Xho* I restriction sites. The obtained recombinant vector pET28a-Ghlac-WT was transformed into *E. coli* BL21 (DE3). The cells containing the recombinant vector were grown in Luria–Bertani (LB) medium supplemented with 25 mg/L of kanamycin. When OD_600_ reached 0.6, 0.5 mM IPTG and 0.5 mM CuSO_4_ were added to induce the expression of Ghlac, and then the cells continued to grow at 16 °C for 16 h. The cells were collected by centrifugation at 4000× *g* for 30 min and homogenized using a JN-Mini homogenizer (JNBio, Guangzhou, China). The recombinant Ghlac in the supernatant was purified using Ni-NTA resin according to the reported method [50]. The purified Ghlac in 20 mM phosphate buffer (pH 7.4) was stored at −80 °C. The purity and molecular mass of Ghlac were assessed by SDS-PAGE. The UV/visible absorption spectrum of Ghlac was scanned in the range of 200–800 nm using a SpectraMax M2e Microplate Reader (Molecular Devices, Sunnyvale, CA, USA). The copper content of Ghlac was analyzed with an iCAP Qc inductively coupled plasma mass spectrometry (ICP-MS) (ThermoFisher Scientific, Waltham, MA, USA) [22].

### 3.3. Mutation Design Using PROSS and Site-Directed Mutagenesis

Goldenzweig et al. developed an automated structure- and sequence-based algorithm, the Protein Repair One Stop Shop (PROSS) webserver, to design protein variants with enhanced stability requiring minimal experimental testing (accessed on 18 May 2020) [26]. Ghlac sequence was submitted to PROSS with N41, H78, C119, and H136 constrained to improve the thermostability. The designed variants with 17, 25, and 31 mutated residues (referred to as Ghlac Mut1, Mut2, and Mut3, respectively; Appendix A) were chosen to test according to the manual of PROSS.

The variants H78A, C119A, and H136A, as well as the combination of H78A, C119A, and H136A (referred to as 3A), of Ghlac Mut2 were constructed using the one step site-directed mutagenesis method. Briefly, the primers with the desired mutation were designed and synthesized (Appendix A). PCR was performed with the plasmid pET28a-Ghlac-Mut2 as the template. The PCR products were digested with *Dpn* I at 37 °C for 4 h and transformed into DH5α competent cells. The variants were confirmed by sequencing, expressed, and purified according to the method described in Section 3.2.

### 3.4. Enzyme Activity Assay

The activity of Ghlac was determined with ABTS (ε_420_ = 38,000 M^−1^ cm^−1^) as the substrate. The reaction system contained appropriately diluted purified Ghlac, 1 mM ABTS, 1 mM CuSO_4_, and 20 mM acetic acid-sodium acetate buffer (pH 4.0). The reaction samples were incubated at 50 °C for 10 min and then put on ice for 5 min to terminate the reaction. The absorbance at 420 nm was measured. The laccase activity against 1 mM DMP (ε_468_ = 35,640 M^−1^ cm^−1^), 1 mM guaiacol (ε_470_ = 26,600 M^−1^ cm^−1^), and 25 μM SGZ (ε_530_ = 64,000 M^−1^ cm^−1^) was also determined. One unit of enzyme activity (1 U) was defined as the amount of enzyme required to oxidize 1 μmol of substrate per minute at 50 °C. The kinetic parameters (*K*_m_ and *k*_cat_) of Ghlac toward ABTS were determined at 60 °C according to the standard laccase assay method.

### 3.5. Effects of Temperature and pH on the Laccase Activity and Stability

The effect of pH on the laccase activity was determined using 20 mM acetic acid-sodium acetate (pH 3.0–6.0) and 20 mM Tris-HCl (pH 7.0–8.0). To evaluate its pH stability, Ghlac was incubated in different pH buffers (3.0–8.0) at 4 °C for 6 h, and the residual activity was measured according to the standard laccase assay method. The effect of temperature on the laccase activity was analyzed in 20 mM acetic acid-sodium acetate buffer (pH 4.0) at temperatures ranging from 30 to 80 °C. To assess the thermostability, Ghlac was incubated at 50 °C and 60 °C in 20 mM phosphate buffer (pH 7.4) for various time intervals, and the residual activity was measured. The Thermofluor assay was performed using a 96-well Applied Biosystems QuantStudio7Flex qPCR instrument (ThermoFisher Scientific, Waltham, MA, USA). The mixture containing 0.3 mg/mL of Ghlac, 5 × Sypro orange, and 100 mM citrate buffer (pH 4.0) was incubated for 10 min on ice before the heating process. The temperature was programmed to increase from 25 to 95 °C at a rate of 1 °C/min in a step-and-hold manner, and the fluorescence in the x1-m2 channel was recorded. The melting temperature (T_m_) was calculated by fitting the fluorescence data to the Boltzmann equation. To examine the effect of copper ions on the thermostability of Ghlac, CuSO_4_ was added to the mixture, and T_m_ of Ghlac with Cu^2+^ was determined.

### 3.6. Effects of Metal Ions and Organic Solvents on the Activity of Mut2

Different concentrations of metal ions (1 mM and 10 mM) and organic solvents (10% and 20%, *v*/*v*) were added to the reaction system to study their effects on the activity of Ghlac Mut2. After pre-incubation of Mut2 with metal ions (Na^+^, Mg^2+^, Ca^2+^, Mn^2+^, Ni^2+^, Zn^2+^, Ba^2+^, and EDTA) and organic solvents (formaldehyde, methanol, ethanol, acetone, isopropanol, and dimethyl sulfoxide) at 4 °C for 30 min, the laccase activity was determined according to the standard laccase assay method.

### 3.7. MG Decolorization Catalyzed by Mut2

The primary decolorization mixture contained 40 U/L of Mut2, 100 mg/L of MG, 1 mM CuSO_4_, and 20 mM acetic acid-sodium acetate buffer (pH 4.0). Decolorization was performed at 60 °C for 2 h in the dark without shaking. The absorbance at 617 nm was measured to detect MG decolorization using Formula (1):Decolorization rate (%) = (*A*_0_ − *A*)/*A*_0_ × 100(1)
where *A*_0_ is the absorbance of untreated MG and *A* is the absorbance of MG treated by Mut2.

The optimal condition of MG decolorization catalyzed by Mut2 was determined. The effect of mediators on MG decolorization was analyzed by adding 0.01 mM of different mediators (ABTS, HBT, VA, MeS, and ASG) to the primary decolorization mixture. The concentration of ABTS (0.01–0.4 mM) was then optimized. The optimal pH was determined in different pH buffers (pH 3.0–5.5) with 0.1 mM ABTS. The optimal temperature was determined at 45 to 85 °C in the pH 4.0 buffer containing 0.1 mM ABTS. The time course of MG decolorization catalyzed by Mut2 in the presence of ABTS was recorded in the pH 4.0 buffer at 70 °C for 3 h.

### 3.8. Toxicity Tests

The bacteria (*E. coli* and *B. subtilis*) and plant (*Z. mays*) seeds were used to evaluate the toxicity of 100 mg/L of MG before and after treatment by Mut2 at 70 °C for 3 h. The pH of the samples was adjusted to 7.0 to eliminate the effect of pH on the growth of bacteria and plant seeds.

The toxicity to bacteria was assessed using the gram-negative bacteria *E. coli* and the gram-positive bacteria *B. subtilis* [51]. The samples (4.0 mL) before and after Mut2 treatment were mixed with 5 × LB medium (1.0 mL). The bacteria were inoculated to the mixed solution and incubated at 37 °C. The bacteria growing in 1 × LB medium diluted from 5 × LB medium with distilled water were set as the control in parallel. The absorbance at 600 nm was measured to detect the growth of bacteria.

The phytotoxicity assay using *Z. mays* seeds was performed according to the previous method [52]. The samples before and after Mut2 treatment were added to the Petri dishes containing the double-layered filter paper. The seeds were placed in the Petri dish and incubated at 25 °C for 4 d in the dark. The seeds growing in distilled water were set as the control in parallel. The lengths of the plumule and radicle were recorded, and the germination rate was calculated according to the following Formula (2):Germination rate (%) = *n*/*N* × 100(2)
where *n* is the number of germinated seeds and *N* is the number of total tested seeds.

### 3.9. Sequence Analysis

The characteristics of Ghlac (molecular weight and pI) were determined using the ProtParam tool available on the Expasy server (https://web.expasy.org/protparam/, accessed on 15 June 2019). Multiple sequence alignment was performed using Clustal Omega (accessed on 18 May 2020) [53]. The structural model of Ghlac was constructed using SWISS-MODEL (accessed on 18 May 2020) [54] and analyzed using Pymol [55].

### 3.10. Statistical Analysis

All experiments were performed at least three times, and the data were presented as mean ± standard deviation (SD). Microsoft Excel version 2016 (Microsoft Corporation, Redmond, WA, USA) and GraphPad Prism 8.0 (San Diego, CA, USA) were used for all statistical analysis.

## 4. Conclusions

Ghlac, a novel member of the DUF152 family, was cloned from *G. hydrogeniphilus* and characterized as a functional laccase. By the structural and biochemical analyses, the conserved residues H78, C119, and H136 were identified to form the putative copper binding site. In addition, the thermostable Ghlac variant Mut2 was highly tolerant to alkaline conditions and organic solvents. Furthermore, Mut2 could efficiently decolorize MG and thoroughly eliminate the toxicity of MG in the presence of ABTS at high temperatures, showing great potential to remediate MG effluent immediately discharged from the dyeing process. However, further studies on the catalytic mechanism of Mut2 and co-immobilization of laccase and mediator need to be done to facilitate its industrial application.

## Figures and Tables

**Figure 1 ijms-22-11755-f001:**
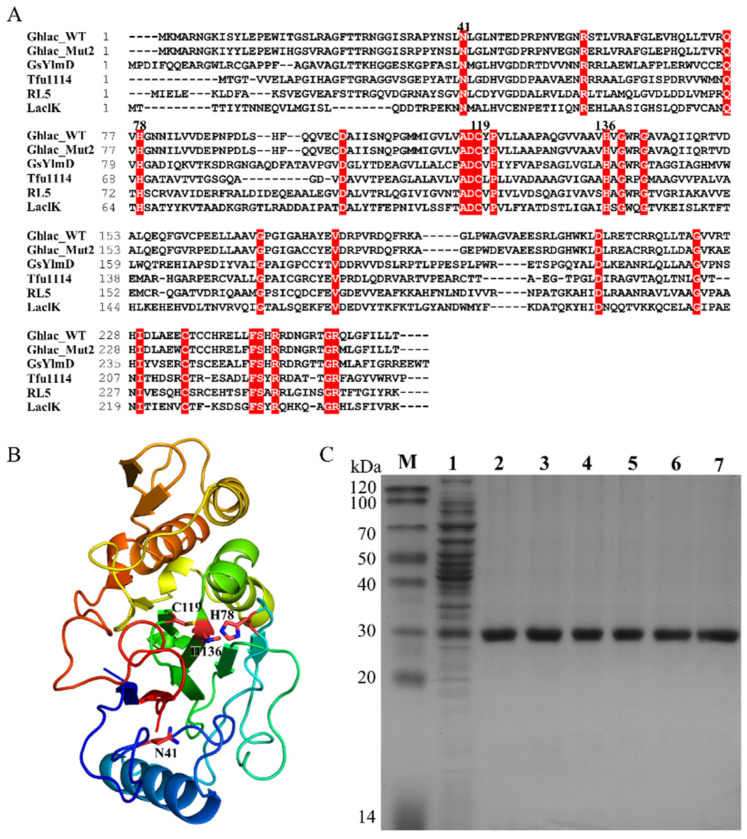
Structure analysis and purification of Ghlac. (**A**) The sequence alignment of Ghlac WT, Mut2, GsYlmD (WP_053413740.1), RL5 (CAK32504.1), Tfu1114 (AAZ55152.1), and LaclK (WP_029500662). The identical residues are highlighted in red. The putative residues binding with copper ions are labeled with the residue numbers. (**B**) Structural model of Ghlac based on GsYlmd (PDB: 6T0Y, https://www.rcsb.org/structure/6T0Y, accessed on 18 May 2020). The putative residues binding with copper ions are labeled. (**C**) SDS-PAGE analysis of Ghlac purified using Ni-NTA chromatography. Lane M: protein marker; lane 1: the supernatant of the homogenized cells expressing Ghlac WT; lane 2–7: the purified Ghlac WT, Mut2, H78A, C119A, H136A, and 3A, respectively.

**Figure 2 ijms-22-11755-f002:**
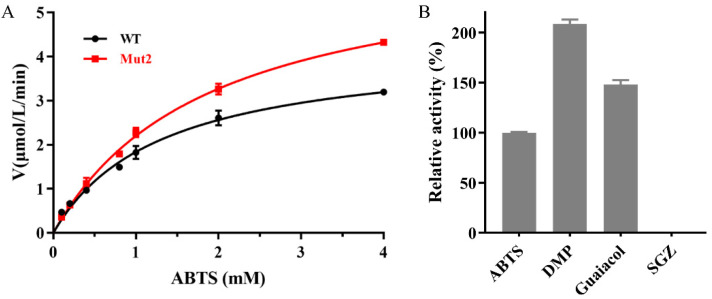
Kinetic analysis of Ghlac (**A**) and substrate specificity of Mut2 (**B**). ABTS, DMP, and SGZ were the abbreviations of 2,2′-azino-bis(3-ethylbenzthiazoline)-6-sulfonate, 2,6-dimethoxyphenol, and syringaldazine, respectively. *K*_m_ and *k*_cat_ of Ghlac WT were 1.3 mM and 125.7 min^−1^ (4.1 U/mg), respectively. *K*_m_ and *k*_cat_ of Mut2 were 1.9 mM and 188.9 min^−1^ (6.2 U/mg), respectively.

**Figure 3 ijms-22-11755-f003:**
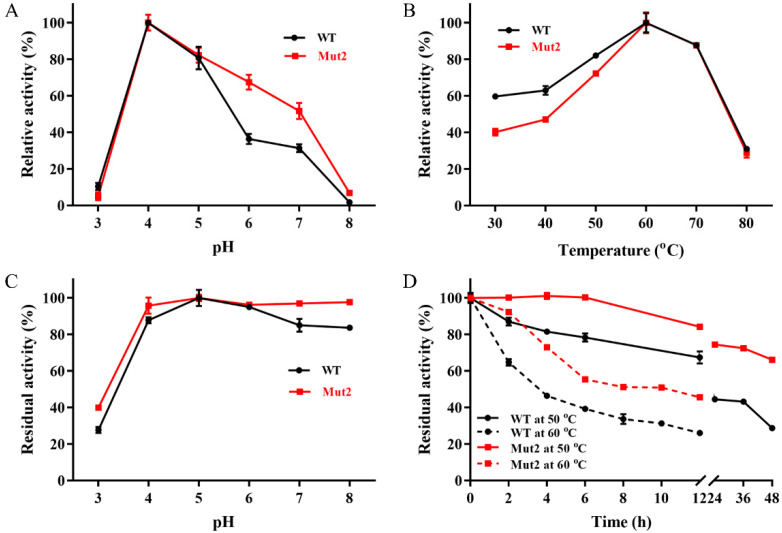
Effects of pH and temperature on the activity and stability of Ghlac. (**A**) The optimal pH (**A**) and temperature (**B**) for Ghlac. Effect of pH (**C**) and temperature (**D**) on the stability of Ghlac. The t_1/2_ values of WT and Mut2 at 50 °C were 21.9 h and 80.6 h, respectively. The t_1/2_ values of WT and Mut2 at 60 °C were 3.6 h and 9.8 h, respectively.

**Figure 4 ijms-22-11755-f004:**
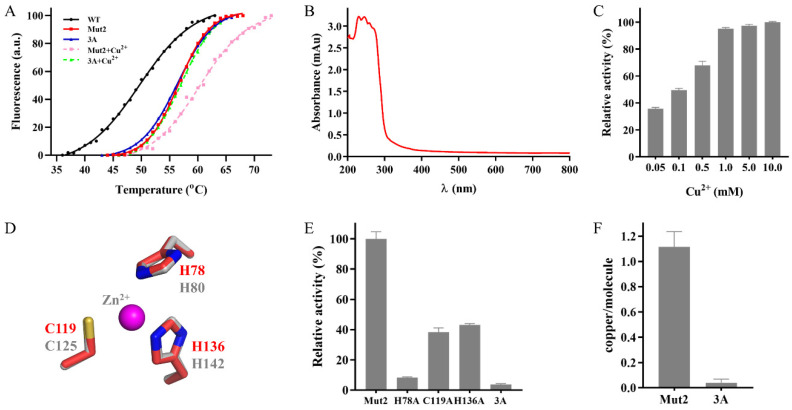
Identification of the putative copper binding site. (**A**) The effect of Cu^2+^ on the thermostability of Ghlac using the Thermofluor assay. The T_m_ values of Ghlac WT, Mut2, 3A, Mut2 with Cu^2+^ (Mut2+Cu^2+^), and 3A with Cu^2+^ (3A+Cu^2+^) were 49.5 ± 0.2 °C, 56.4 ± 0.2 °C, 56.4 ± 0.1 °C, 60.9 ± 0.6 °C, and 56.2 ± 0.1 °C, respectively. (**B**) The UV/visible spectrum of Mut2. (**C**) The effect of Cu^2+^ on the activity of Mut2. (**D**) Superposition of the putative copper binding sites of Mut2 and GsYlmd (PDB: 6T0Y). The residues of Mut2 and GsYlmd were labeled in red and grey, respectively. Zn^2+^ in GsYlmd was shown as a magenta sphere. (**E**) The laccase activity of Mut2 and its variants. (**F**) The copper content of Mut2 and the variant 3A determined by ICP-MS.

**Figure 5 ijms-22-11755-f005:**
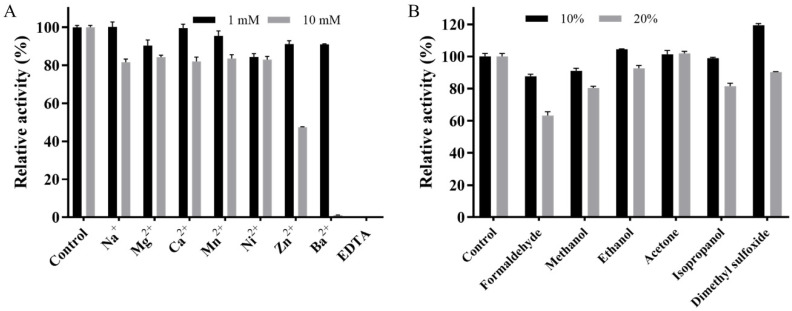
Effects of metal ions (**A**) and organic solvents (**B**) on the activity of Mut2.

**Figure 6 ijms-22-11755-f006:**
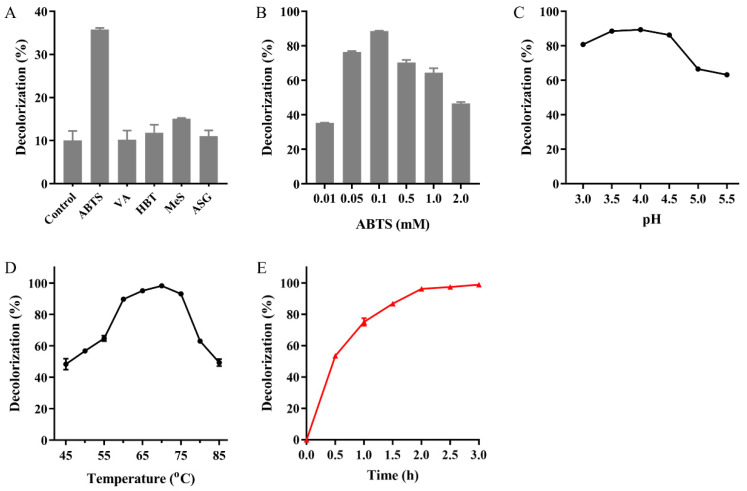
MG decolorization catalyzed by Mut2. The effects of different mediators (**A**) and ABTS concentration (**B**) on MG decolorization. VA, HBT, MeS, and ASG were the abbreviations of violuric acid, 1-hydroxybenzotriazole, Methyl syringate, and acetosyringone, respectively. The optimal pH (**C**) and temperature (**D**) of MG decolorization. (**E**) The time course of MG decolorization.

**Figure 7 ijms-22-11755-f007:**
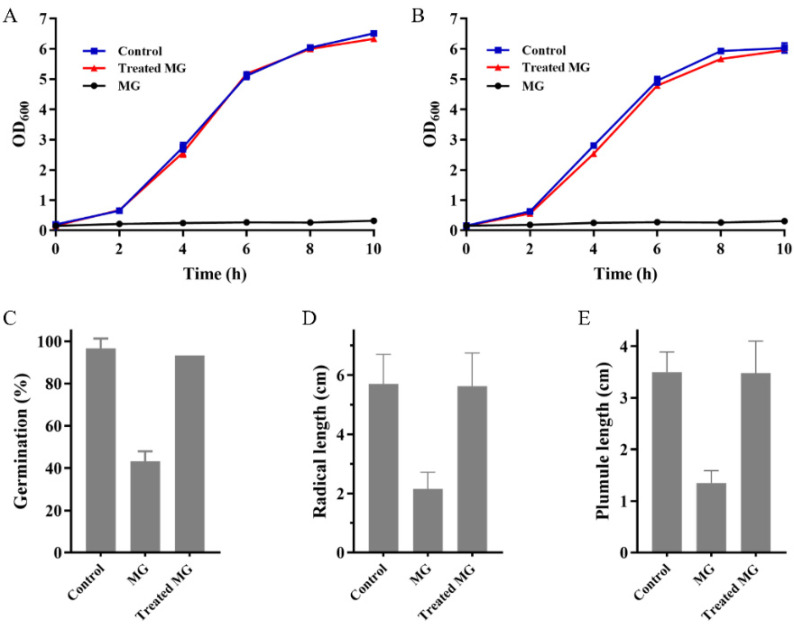
Toxicity evaluation of MG before and after Mut2 treatment. The growth curve of *E. coli* (**A**) and *B. subtilis* (**B**) in LB medium containing MG and treated MG. (**C**) The germination rate of *Z. mays* seeds in the solution of MG and treated MG. The radical length (**D**) and plumule length (**E**) of germinated *Z. mays* in the solution of MG and treated MG.

**Table 1 ijms-22-11755-t001:** Comparison of the thermostability and MG decolorization ability of bacterial laccases.

Laccase	Source	t_1/2_		MG Decolorization				
		50 °C	60 °C	MG (mg/L)	Mediator	Temperature (°C)	Time (h)	Decolorization Rate
Ghlac Mut2	*G. hydrogeniphilus*	80.6 h	9.8 h	100	0.1 mM ABTS	70	3	>99%
						60		>90%
CotA WLF [17]	*B. pumilus*	6.5 h	ND	50	1 mM ASG	37	10	>95%
pLac*_Si_* [18]	*S. indolifex*	Unstable ^a^	Unstable ^b^	50	1 mM ABTS	30	overnight	>80
FNTL [19]	*Bacillus* sp.	ND	About 2.7 h	50	2 mM ASG	40	0.5	>99%
LaclK [24]	*K. huakuii*	ND	Highly stable ^c^	9	0.1 mM ABTS	60	1	<40%
BaCotA [27]	*B. stratosphericus*	2 h	1 h	100	0.01 mM ABTS	60	3	82%
rLac [31]	*K. pneumoniae*	Stable ^d^	5 h	100		70	1.5	90%
rLAC [46]	*B. amyloliquefaciens*	Stable ^e^	Stable ^f^	100	0.1 mM ABTS	60	6	95%
CueO-p [49]	*E. coli*	Stable ^g^		80	0.1 mM ASG	55	12	98.5%

^a^ and ^b^: pLac*_Si_* retained less than 10% of its activity for 6 h at the indicated temperature; ^c^: LaclK retained over 80% of its activity for 144 h at 60 °C. ^d^: rLac retained more about 60% of its activity after 5 h incubation at 50 °C and retained about 35% of its activity at 70 °C for 1 h. ^e^ and ^f^: rLAC retained 73% and 63% of original activity for 2 h at the indicated temperature, respectively. ^g^: CueO-p retained about 80% of its activity after 4 h incubation at 55 °C. ND: not determined.

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
