# Peer review of "An Engineered Thermostable Laccase with Great Ability to Decolorize and Detoxify Malachite Green"

_ijms, 2021, doi:10.3390/ijms222111755_

Round 1

Reviewer 1 Report

The manuscript by Mao et al. “An Engineered Thermostable Laccase with Great Ability to De- 2 colorize and Detoxify Malachite Green” is interesting and needs minor changes before its publications as follows:

Comments.

  1. Lines 35-38, the authors may elaborate this sentence on each application with related citations examples i.e. delignification to improve biomass saccharification (doi: 10.1002/biot.201800468).
  2. Lines 39-41, elaborate the sentence “bacterial laccases exhibit quite low redox potential ~ 400 mV as compared to fungal laccases with higher redox potential between 470 and 810” i.e. doi: 10.1007/s12088-020-00912-4.
  3. Line 49, how about concentration or discharge of MG as industrial effluents?
  4. Figure 1 and Figure 2 may be merged.
  5. The authors should follow the significant figures rule for the presentation of data in the text.
  6. Fig. 3, in general, the laccases showed lower activity towards DMP as compared to ABTS. Why this laccase shows higher selectivity towards DMP over ABTS. Please explain it with evidence. How about others WT, mut1, and mut3? Also, possible explanation and discussion for higher activity over WT with more experimental inside like structural changes, etc. 
  7. Why activity in Mut1 and Mut3 is lower than WT? Have any specific amino acids residue involves in the catalysis mechanism as a major determinant? The results may be validated i.e. doi: 10.3390/ijms21217859. 
  8. How about the specific activity of WT and various mutants? It will be better to show the actual specific activity in Figure 4.
  9. In figure 7, the authors may provide the data of WT as control.
  10. The authors should avoid the uses too many old references before 2016. Mostly delete them or replace them with recent citations.

Reviewer 2 Report

The manuscript entitled „An engineered thermostable laccase with great ability to decolorize and detoxify malachite green” deals with the receipt of recombinant laccase and its properties and application to dye detoxicity and decolorization. In my opinion the data presented in this manuscript supplement the current knowledge and are in line with journal scope. The manuscript is interesting, is well written but there are some editorial shortcomings which should have been corrected before publication of the manuscript. Besides, the Authors should pay more attention to the questions below, what in my opinion will makes the manuscript more interesting. In conclusion, he finds this manuscript suitable for publication after minor revisions.

My questions and comments after reading of the manuscript are:

Q1. According to my expirience it is rather untypical the lack of activity of laccase for syryngaldazine. Could you explain the lack of activity of studied laccase with syryngaldazine? Maybe do you know some other examples of such a laccases?

Q2. Could you try explain the reason of absorbance quenching at 610 nm for studied laccase?

Q3.  What is the role of laccase in anaerobe organism since the activity of laccases are connected with oxygen reduction?

Q4. Is there a possibility that this protein is not laccase after all?

I ask the Authors to address these questions and to include the answers if they see fit, in the text of the manuscript.

Editorial remarks:

Abstract:

line 16 is novol should be rather novel

Introduction:

line 45 -47 Do you compare bacterial and fungal laccases? But what is better? because you write only about bacterial laccases in this sentence.

Results and Discussion:

line: 131 Is the reference to Figure 4B here correct? Should it not be a reference to another figure?

Materials and Methods:

confused section numbering, there are almost only number 3.2 as paragraphs during section

Round 2

Reviewer 1 Report

Accept as is